# Integrated Design of Aerodynamic Performance and Structural Characteristics for Medium Thickness Wind Turbine Airfoil

**Quan Wang \*, Pan Huang \*, Di Gan and Jun Wang**

School of Mechanical Engineering, Hubei University of Technology, Wuhan 430068, China;
gandi10086@126.com (D.G.); quan_wang@foxmail.com (J.W.)

**\*** Correspondence: wangquan2013@hbut.edu.cn (Q.W.); 101810086@hbut.edu.cn (P.H.)

**Abstract:** The currently geometric and aerodynamic characteristics for wind turbine airfoils with the medium thickness are studied to pursue maximum aerodynamic performance, while the interaction between blade stiffness and aerodynamic performance is neglected. Combining the airfoil functional integration theory and the mathematical model of the blade cross-section stiffness matrix, an integrated design method of aerodynamic performance and structural stiffness characteristics for the medium thickness airfoils is presented. The aerodynamic and structural comparison of the optimized WQ-A300 airfoil, WQ-B300 airfoil, and the classic DU97-W-300 airfoil were analyzed. The results show that the aerodynamic performance of the WQ-A300 and WQ-B300 airfoils are better than that of the DU97-W-300 airfoil. Though the aerodynamic performance of the WQ-B300 airfoil is slightly reduced compared to the WQ-A300 airfoil, its blade cross-sectional stiffness properties are improved as the flapwise and edgewise stiffness are increased by 6.2% and 8.4%, respectively. This study verifies the feasibility for the novel design method. Moreover, it also provides a good design idea for the wind turbine airfoils and blade structural properties with medium or large thickness.

**Keywords:** medium thickness airfoils; aerodynamic performance; structural stiffness; ply parameters; integrated design

## 1. Introduction

Wind energy, as a clean and renewable energy, has attracted more and more attention from all over the world. Wind turbines that can transfer wind energy into electricity production have been developing rapidly in the world in recent years. The blades, as one of the key components of wind turbines, cost about 20% of the whole machine. Their good design, reliable quality, and superior performance are the decisive factors to improve the utilization rate of wind energy and ensure stable operation of wind turbines. The key factors for the design of wind turbine blades are the aerodynamic performance and the structural properties of wind turbine airfoils.

At present, the design of airfoil profiles with medium thickness, in which the maximum relative thickness is between 25% and 35%, has been the focus of aerodynamic performance of wind turbine blades. Tangler and Somers [1,2] designed 35 kinds of airfoils named the NREL-S (National Renewable Energy Laboratory Special) airfoil series for various wind turbines using Eppler theory and inverse design method. The new airfoil families exhibit high maximum lift coefficients. DU airfoil families with a relative thickness of 15% to 40% were designed using mixed inverse design method [3,4]. Airfoils with good aerodynamic performance can be designed by changing the upper-surface thickness and the S-shape lower surface. The result showed that the lift coefficient was improved and relatively insensitive to roughness. Compared with other airfoils, the DU (Delft University) airfoil families exhibited better

aerodynamic performance. From the middle of 1990's, RISØ airfoils were designed by RISØ National Laboratory in Denmark [5]. In their work a direct design method for wind turbine airfoils based on mathematic optimization and the XFOIL software was presented. Sobieczky [6] presented a method named PARSEC (Pseudopotential Algorithm for Real-Space Electronic Calculations) to design airfoils; a modern airfoil was designed by controlling the airfoil geometry parameters. Hajek [7] presented an improved PARSEC method to design airfoils, and a brand-new airfoil was optimized. Ava and Alireza [8] presented a new method for airfoil shape parameterization to optimize the airfoil at high Reynolds number turbulent flow conditions using a genetic algorithm. It was concluded that this method is capable of finding efficient and optimum airfoils in fewer generations. Chen and Wang [9,10] presented a general integral expression of airfoils based on a generalized functional and Trajkovski conformal transformation. CQU-DTU (Chongqing University) and WT (Wind Turbine) series airfoils have been designed successively using a multi-disciplinary optimal method. The aerodynamic performance of the airfoil series has been verified by wind tunnel experiments. Seong-Ho Seo and Cheol-Hyun Hong [11] studied performance improvement of airfoils for wind turbine blades with groove. It was confirmed that the shape of the groove contributed to recovering velocity around the airfoil wall, and the lift to drag the ratio improvement by the groove was maintained at the given range of Reynolds number. Cheng and Zhu [12] used this method combined with airfoil self-noise theory, as such CQU-DTU-LN1 airfoils were designed which exhibited low noise emissions. The noise characteristics of the airfoils were verified by wind tunnel experiments. Zhu and Shen [13] presented an integrated optimal method of airfoil and blade for large wind turbines. The results indicated that the airfoils achieved a high power coefficient and were insensitive to surface roughness. Seyed Mehdi Mortazavi [14] presented a multi-objective genetic algorithm with objective functions of minimum energy waste and maximum efficiency. The results show that using the second law approach along with the Pareto optimality concept leads to a set of precise solutions. Xingxing Li [15,16] put forward a mathematical model of the overall optimization employing airfoil performance evaluation indicators. Based on this model, an integrated optimization platform for thick airfoils was established. The results confirmed that the proposed method effectively balanced airfoil's complicated requirements and successfully improved the new airfoil's overall performance. Pierluigi [17] presented a novel optimal procedure for airfoil profile design based on PARSEC parameterization and genetic algorithm optimization. As a matter of fact, the optimization under Nash equilibria solutions would be more attractive to use when a well posed distinction between player variables exists. Pereira and Timmer [18] described a design method of airfoils which were suitable to employ actuation in a wind energy environment. The novel airfoil sections are baptized wind energy actuated profiles. The results show that using WAP (wind energy actuated profiles) airfoils provided much higher control efficiency than adding actuation on reference wind energy airfoils, without detrimental effects in non-actuated operation. Miller [19] used a Bézier curve and XFOIL software to design flatback airfoil families with a high degree of attention to the appropriate selection of design constraints and objectives. The new CU-W1-XX series were shown to have equal or superior performance compared to other airfoils. Ram [20] designed the airfoil sections named USP07-45XX for a 20 kw wind turbine using the multi-objective genetic algorithm. The USP07-45XX airfoils showed only a slight change during clean and soiled conditions both in experiments and in numerical studies. Besides, there are many other researchers [21–26] who have also made other significant contributions to this field. In summary, some new design methods were put forward to optimize wind turbine airfoils from different points of view. Those new wind turbine airfoils not only have improved the aerodynamic performance, but also increased the efficiency of wind energy.

However, for the above studies, whether they were based on an original airfoil profiles, or a mathematical parametrical expression, the objective functions usually were the aerodynamic parameters (such as lift-drag ratio) of the free transition and the fixed transition. The influence between blade stiffness and aerodynamic geometry was not considered. Medium thickness wind turbine airfoils, not only require high aerodynamic performance to obtain high wind energy utilization,

but also large structural stiffness characteristics to resist excessive elastic deformation of the blade. Therefore, an optimal design method for the medium thickness wind turbine airfoils in which the aerodynamic performance and blade cross-sectional stiffness is both considered are proposed in this paper. It does not improve the airfoil's aerodynamic performance unilaterally, but seeks to increase both aerodynamic performance and cross-sectional stiffness of the medium thickness airfoils. An optimal mathematic model which combines the functional integrated expression of airfoils and the stiffness matrix of the composite wind turbine blade cross-section is established. Coupled with RFOIL software [27] and the blade cross-sectional stiffness calculated program, the aerodynamic performance of the medium thickness airfoil and the stiffness performance of its cross-section are optimized using a genetic algorithm. Lastly, the optimized airfoil named WQ-B300 is compared with the classic DU97-W-300 airfoil and the WQ-A300 airfoil, which were designed without considering the change of the blade layer parameters, to verify that the airfoil has better aerodynamic performance and structural stiffness characteristics.

## 2. Airfoils Functional Integrated Express

A circle in the $Z$ plane can be translated into a curve resembling an airfoil profile in the $\zeta$ plane by a conformal transformation [28,29], as shown in Figure 1. Where $\zeta$ is a complex quantity defining the points in the plane describing the geometry of the airfoil, $z_c$ is another complex quantity defining the shape points in the plane describing the circular curve and the geometrical scale. Factor $a$ stands for a dimension length.

If the inverse transformation is used to an arbitrary airfoil, the curve in the $Z$ plane will be nearly circular. Because of limited the length of the article, the relevant theoretical knowledge can be read in literature [9,10].

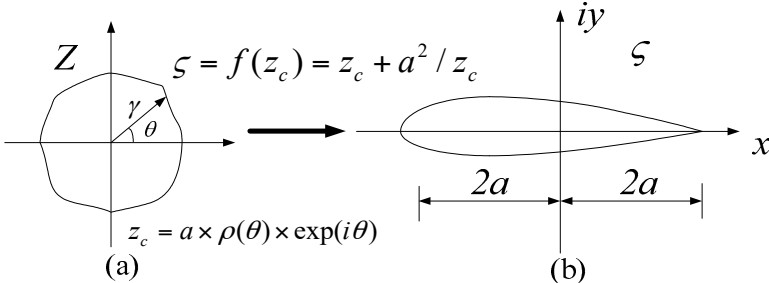

**Figure 1.** Conformal transformation.

The coordinate data of an arbitrary airfoil in the $\zeta$ plane can be written as follows:

$$\begin{cases} x = \left(r + \frac{a^2}{r}\right)\cos\theta \\ y = \left(r - \frac{a^2}{r}\right)\sin\theta \end{cases} \tag{1}$$

where $x$, $y$ are the abscissa and the ordinate of an airfoil, respectively; $r = a\,\exp(\varphi(\theta))$ is the radius vector of an airfoil in the $Z$ plane; $a$ is the geometrical scale factor; and $\theta$ is the argument.

If $\varphi(\theta)$ is a constant, the above expression is a circle. As a result, the transformed geometry is a symmetrical airfoil without chamber. If $\varphi(\theta)$ is a variable unknown function, many different unsymmetrical airfoils with chamber can be transformed by taking different $\varphi(\theta)$.

Based on the Taylor series theory, the trigonometric series is used to express the functional integral equation $\varphi(\theta)$:

$$\varphi(\theta) = a_1(1 - \cos\theta) + b_1\sin\theta + a_2(1 - \cos\theta)^2 + b_2\sin\theta^2 + \ldots$$
$$+ a_k(1 - \cos\theta)^k + b_k\sin\theta^k + \ldots \quad (k = 1, 2, \ldots, n) \tag{2}$$

The mathematical model for the wind turbine airfoil profiles with the medium thickness can be expressed by Equations (1) and (2).

## 3. Stiffness Matrix Calculation of Composite Blade Section

The cross-sectional structure of the wind turbine blade is shown in Figure 2. It is mainly composed of the blade leading edge, main beam, trailing edge, and web. Each part of the blade section is laid with different composite laminates. Using the idea of discretization and equivalent stiffness superposition [30,31], the blade section is discretized into several regional segments along the wind turbine airfoil profile direction. Then, combined with the theory of composite laminates, the effective engineering constants of each layer of the segment in the region are calculated. After that, the equivalent characteristics of the regional segments are obtained by the weighting method. Finally, the blade cross-sectional structure characteristics of each segment are obtained by summation. Therefore, the stiffness characteristics of the blade cross-section are obtained. The flow chart of the model is shown in Figure 3.

The aerodynamic shape of the wind turbine airfoils and the blade structural parameters are particularly important in the flow chart. The aerodynamic performance of the blade is determined directly by the wind turbine airfoil's geometry. The stiffness characteristics of the blade are directly determined by the internal plying parameters (such as the layer thickness, layer angle, layer sequence and so on). After the blade cross-section is discretized, each segment is laid by several laminates. The effective engineering constants of the ply can be obtained by Equations (3) and (4), when the layer angle of the unidirectional ply is $\alpha$.

$$E_x^{ply} = \frac{1}{\frac{1}{E_1}\cos^4\alpha + \left(\frac{1}{G_{12}} - \frac{2v_{12}}{E_1}\right)\sin^2\alpha\cos^2\alpha + \frac{1}{E_2}\sin^4\alpha} \tag{3}$$

$$G_{xy}^{ply} = \frac{1}{\left[\frac{4}{E_2} + \frac{4+8v_{12}}{E_1} - \frac{2}{G_{12}}\right]\sin^2\alpha\cos^2\alpha + \frac{\sin^4\alpha + \cos^4\alpha}{G_{12}}} \tag{4}$$

where $E_x^{ply}$ and $G_{xy}^{ply}$ are the effective Young's modulus and shear modulus with a layer angle, respectively; $E_1$ and $E_2$ are the elastic modulus of materials; $G_{12}$ is shear modulus of materials; and $v_{12}$ is Poisson's ratio.

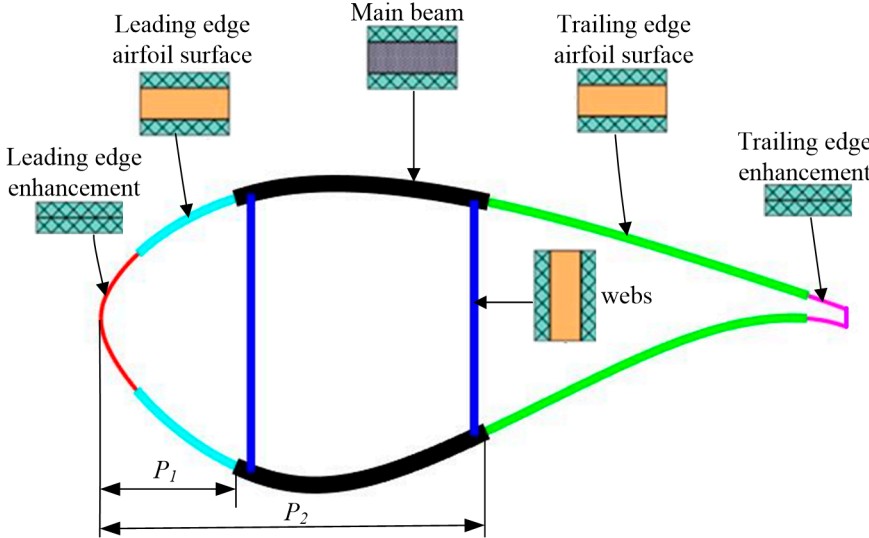

**Figure 2.** Schematic diagrams of the blade cross-sectional parameters.

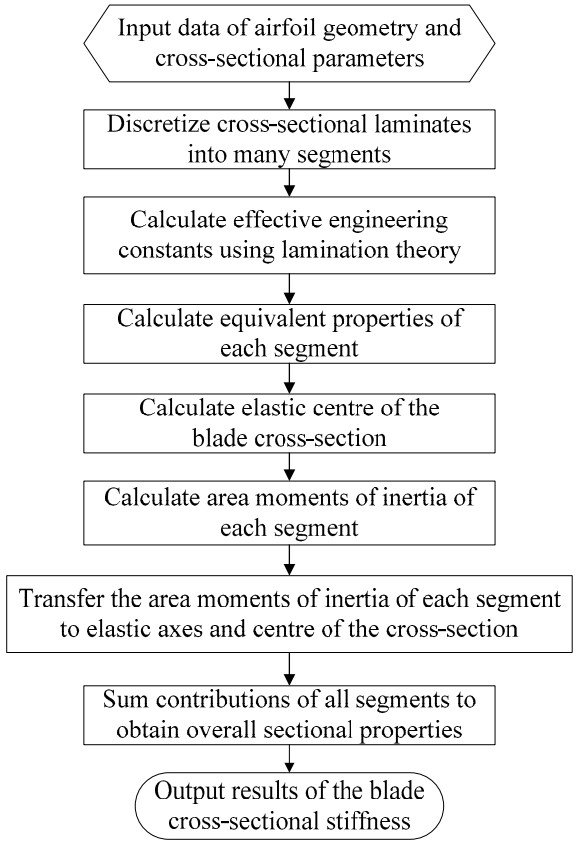

**Figure 3.** Flow chart of the mathematical model.

As is shown in Figure 4, the equivalent Young's modulus $E_{equ}$, shear modulus, thickness $t_{equ}$, and $A_{equ}$ area can be expressed as:

$$E_{equ} = \frac{\sum_{i=1}^{m} E_{x,i}^{ply} t_i^{ply}}{\sum_{i=1}^{m} t_i^{ply}} \tag{5}$$

$$G_{equ} = \frac{\sum_{i=1}^{m} G_{xy,i}^{ply} t_i^{ply}}{\sum_{i=1}^{m} t_i^{ply}} \tag{6}$$

$$t_{equ} = \sum_{i=1}^{m} t_i^{ply} \tag{7}$$

$$A_{equ} = \sum_{i=1}^{m} A_i^{ply} = \sum_{i=1}^{m} t_i^{ply} w_{seg} \tag{8}$$

where $i$ is the *ith* ply in a segment; $m$ indicates the number of plies in an segment; $E_{x,i}^{ply}$ is the effective Young's modulus of the *ith* ply; $t_i^{ply}$ and $A_i^{ply}$ are the thickness and area of the *ith* ply, respectively; and $w_{seg}$ is the width of an segment.

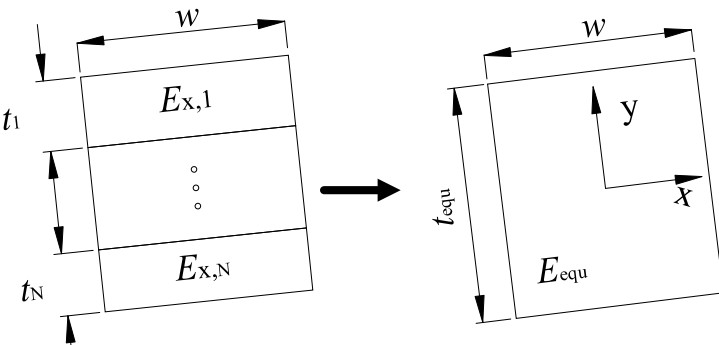

**Figure 4.** Equivalent representation of each segment.

The elastic center of the blade cross-section can also be calculated using the weighting method:

$$X_E = \frac{\sum_{I=1}^{N} E_{equ,I} A_{equ,I} \overline{x}_{c,I}}{\sum_{I=1}^{N} E_{equ,I} A_{equ,I}} \tag{9}$$

$$Y_E = \frac{\sum_{I=1}^{N} E_{equ,I} A_{equ,I} \overline{y}_{c,I}}{\sum_{I=1}^{N} E_{equ,I} A_{equ,I}} \tag{10}$$

where, $I$ is the $I$th segment; $N$ is the number of a segment; $E_{equ,I}$ and $A_{equ,I}$ are the equivalent Young's modulus and area of the $I$ segment, respectively; and $\overline{x}_{c,I}$ and $\overline{y}_{c,I}$ are the mass center coordinates of the $I$th segment.

The moments of inertia for each segment can be calculated using an integral formula:

$$I_{xx} = \int \overline{y}^2 d\overline{x} d\overline{y} \tag{11}$$

$$I_{yy} = \int \overline{x}^2 d\overline{x} d\overline{y} \tag{12}$$

$$I_{xy} = \int \overline{xy} d\overline{x} d\overline{y} \tag{13}$$

where, $I_{xx}$ and $I_{yy}$ are the moment of inertia with respect to $x$ axis and $y$ axis (Figure 4), respectively; and $I_{xy}$ is the product of inertia.

Careful treatment should be given to the above moments of inertia which are calculated relative to the local axes and centroid of each segment. However, the blade cross-sectional properties containing flapwise stiffness and edgewise stiffness are referred to the elastic axes and center of the blade cross-section. Therefore, the moments of inertia in the local axes of each segment can be transferred to the axes which are parallel to the elastic axes of the blade cross-section by using the transform formula:

$$I_{XX} = \frac{I_{xx} + I_{yy}}{2} + \frac{I_{xx} - I_{yy}}{2} \cos 2\beta - I_{xy} \sin 2\beta \tag{14}$$

$$I_{YY} = \frac{I_{xx} + I_{yy}}{2} - \frac{I_{xx} - I_{yy}}{2} \cos 2\beta + I_{xy} \sin 2\beta \tag{15}$$

where, $\beta$ is the angle between the local axes of each segment and the elastic axes of the blade cross-section.

The moments of inertia in Equations (14) and (15) can be transferred to elastic center of the blade cross-section by using the parallel axis theory:

$$I_{XX}^{sec} = I_{XX} + A_{equ}(\overline{x}_c - X_E)^2 \tag{16}$$

$$I_{YY}^{sec} = I_{YY} + A_{equ}\left(\bar{y}_c - Y_E\right)^2 \tag{17}$$

The flapwise stiffness $EI_X$ and edgewise stiffness $EI_Y$ are obtained by summing the segments of the whole blade cross-section:

$$EI_X = \sum_{I=1}^{N} E_{equ,I} I_{XX,I}^{sec} \tag{18}$$

$$EI_Y = \sum_{I=1}^{N} E_{equ,I} I_{YY,I}^{sec} \tag{19}$$

In order to verify the accuracy of the mathematical model of the wind turbine blade cross-sectional stiffness, a 2.1 MW (megawatt) wind turbine blade section, which is located about 40% along the blade span, is taken as an example. The airfoil of this section is the DU97-W-300. The anisotropic material properties [32] of the blade are shown in Table 1. The laminated parameters of the composite blade cross-section are shown in Table 2. The calculated results of this mathematical model are compared with that of the PreComp software [29] and the measured values, as shown in Table 3. From the Table 3, it can be seen that the flapwise and edgewise stiffness calculated by this method are closer to the measured results than those calculated by PreComp software. Compared with the measured stiffness, the flapwise stiffness error is 7.692%, and the edgewise stiffness error is 8.841%. It indicates that this mathematical model can accurately calculate the wind turbine blade cross-sectional stiffness. Moreover, the mathematical model is useful for the optimization of aerodynamic performance and structural stiffness of wind turbine airfoils and blades.

**Table 1.** Material properties of the blade cross-section. PVC (Polyvinyl chloride).

| Num | Materials | $E_1$ (GPa) | $E_2$ (GPa) | $G_{12}$ (GPa) | Poisson Ratio | Density (kg/m$^3$) | Thickness (mm) |
|---|---|---|---|---|---|---|---|
| A | unidirectional | 39.18 | 11.69 | 3.5 | 0.3 | 1940 | 0.892 |
| B | bi-axial | 11.5 | 11.5 | 10.8 | 0.48 | 1970 | 0.555 |
| C | trial laminate | 25.8 | 13.59 | 7.36 | 0.36 | 1842 | 0.906 |
| D | coating | 3.44 | 3.44 | 1.38 | 0.3 | 1235 | 0.6 |
| E | PVC foam | 0.035 | 0.035 | 0.022 | 0.3 | 60 | 5 |

**Table 2.** The structure of blade cross-section plies.

| | Leading Reinforce | Leading Surface | Main Beam | Trailing Surface | Trailing Reinforce | Big Web | Small Web |
|---|---|---|---|---|---|---|---|
| layer | D+2C+22A +2C+D | D+2C+22A+ E+2C+D | D+2B+2C+ 55A+2C+2B+D | D+2C+22A +E+2C+D | D+2C+22A +2C+D | B+2E+B | B+E+B |

**Table 3.** The comparisons of the stiffness performance for a 2.1MW (Megawatt) actual blade cross-section.

| Stiffness | Calculated Results | PreComp Results | Measured Data |
|---|---|---|---|
| $EI_x$(N·m$^2$) | $1.12 \times 10^9$ | $1.21 \times 10^9$ | $1.04 \times 10^9$ |
| $EI_y$(N·m$^2$) | $3.57 \times 10^8$ | $3.64 \times 10^8$ | $3.28 \times 10^8$ |

## 4. The Optimized Mathematical Model

### 4.1. Objective Function

For the medium thickness airfoils, not only the aerodynamic performance but also the blade structural characteristic must be considered. Therefore, the objective function is to maximize the lift-drag ratio and blade cross-sectional stiffness in this study. The aerodynamic performances (such as the lift coefficient and the drag coefficient) are calculated by RFOIL software [3,4] on the condition of smooth and rough at design angle of attack.

$$F(x) = w_1 f_1(x) + w_2 f_2(x) \tag{20}$$

$$f_1(x) = \mu_1 C_L/C_D + \mu_2 C'_L/C'_D \tag{21}$$

$$f_2(x) = \varepsilon_1 EI_X + \varepsilon_2 EI_Y \tag{22}$$

where $C_L/C_D$, $C'_L/C'_D$ indicate the maximized lift-drag ratio on the condition of smooth and rough, respectively; $w_1, w_2$ are the weighting factors for the objective function of aerodynamic performance and the blade cross-sectional stiffness, respectively, $w_1, w_2 \in [0\ 1]$, $w_1 + w_2 = 1$; $\mu_1$, $\mu_2$ are the weighting factors for smooth and rough condition, respectively; $\mu_1$, $\mu_2 \in [0\ 1]$, $\mu_1 + \mu_2 = 1$; $\varepsilon_1$, $\varepsilon_2$ are the weighting factors of flapwise stiffness $EI_X$ and edgewise stiffness $EI_Y$, respectively, $\varepsilon_1, \varepsilon_2 \in [0\ 1]$, $\varepsilon_1 + \varepsilon_2 = 1$.

### 4.2. Design Valuable

On the basis of functional analysis of wind turbine airfoils, it is found that the airfoil shape function $\varphi(\theta)$ directly affects the airfoil geometry characteristics and aerodynamic performance. According to the corresponding constraints, a medium thickness airfoil can be designed to meet the expected aerodynamic performance requirements. By changing the aerodynamic shape of the airfoils, the structural parameters such as moment of inertia, centroid, and stiffness of blade cross-section can be changed. In addition, the player parameters of the blade cross-section directly determine the stiffness characteristics of the blade structure. Therefore, two schemes of the design variables are considered in this paper.

Scheme 1: the coefficients of the shape function $\varphi(\theta)$ from 1 to 6 are chosen as the optimized design variables.

$$X_1 = (a_1, b_1, a_2, b_2, a_3, b_3) \tag{23}$$

Scheme 2: in addition to the first to sixth coefficients of the shape function, the laminated parameters of the blade cross-section at 30% relative thickness along the blade span are also taken as design variables.

$$X_1 = (a_1, b_1, a_2, b_2, a_3, b_3, t_1, t_2, t_3, n_1, n_2, n_3, p_1, p_2) \tag{24}$$

Where $t_1, t_2, t_3$ indicate the variables of plying thickness for the unidirectional layer, biaxial layer, and triaxial layer of the blade cross-section, respectively; $n_1, n_2, n_3$ are the plying number for unidirectional layer, biaxial layer, and triaxial layer of the blade cross-section, respectively; $p_1, p_2$ are the position variables of the blade's main beam (see Figure 2).

### 4.3. Constraint Condition

For case 1, once the design variables are over a certain range of the control points, the profiles would no longer have an airfoil geometry characteristic. So, the design variables are constrained as follows:

$$X_{1min} \leq X_1 \leq X_{1max} \tag{25}$$

For case 2, in addition to constraining the first to sixth coefficients of the airfoil shape function, the laminated parameters of the blade cross-section should also be constrained.

$$X_{2min} \leq X_2 \leq X_{2max} \tag{26}$$

This work is to optimize medium thickness airfoils where the maximum relative thickness is about 30%. Therefore, it is essential to restrict the maximum relative thickness for the design airfoils.

$$\frac{th}{c} = t \in [0.295, 0.305] \tag{27}$$

Geometric compatibility is important for airfoil series of the wind turbine blades. The location of the maximum relative thickness is between 24% and 35% chord:

$$0.24 \leq L_{max} \leq 0.35 \tag{28}$$

One of the objective functions used in this study is the maximal blade cross-sectional flapwise and edgewise stiffness. The larger stiffness will have fewer elastic deflections along the blade. Meanwhile, the cross-sectional mass of the blade should also be limited to the cross-sectional mass of the original blade.

$$(EI_X)_{opt} \geq (EI_X)_{org} \tag{29}$$

$$(EI_Y)_{opt} \geq (EI_Y)_{org} \tag{30}$$

$$mass_{opt} \leq mass_{org} \tag{31}$$

## 5. Optimized Strategies Using a Genetic Algorithm

In order to improve both the aerodynamic performance of the medium thickness airfoil and its blade cross-sectional stiffness, a multi-objective genetic algorithm coupled with the airfoil functional integrated expression and the blade cross-sectional stiffness is introduced to optimize wind turbine airfoils and its structure. The optimal design process is shown in Figure 5. The RFOIL software for calculating the aerodynamic performance of airfoils and the program for calculating the stiffness characteristics of blade cross-section are inserted into the genetic algorithm program to calculate the fitness of the objective function. Then, the parameters of the genetic algorithm are again selected, crossed, and mutated to generate new populations. Finally, the iteration conditions are determined until the new parameters of the optimal airfoils and the blade cross-section are output. The parameters of the genetic algorithm are as follows: Population number is 30, maximum iteration number is 200, crossover operator is 0.7, and mutation operator is 0.2.

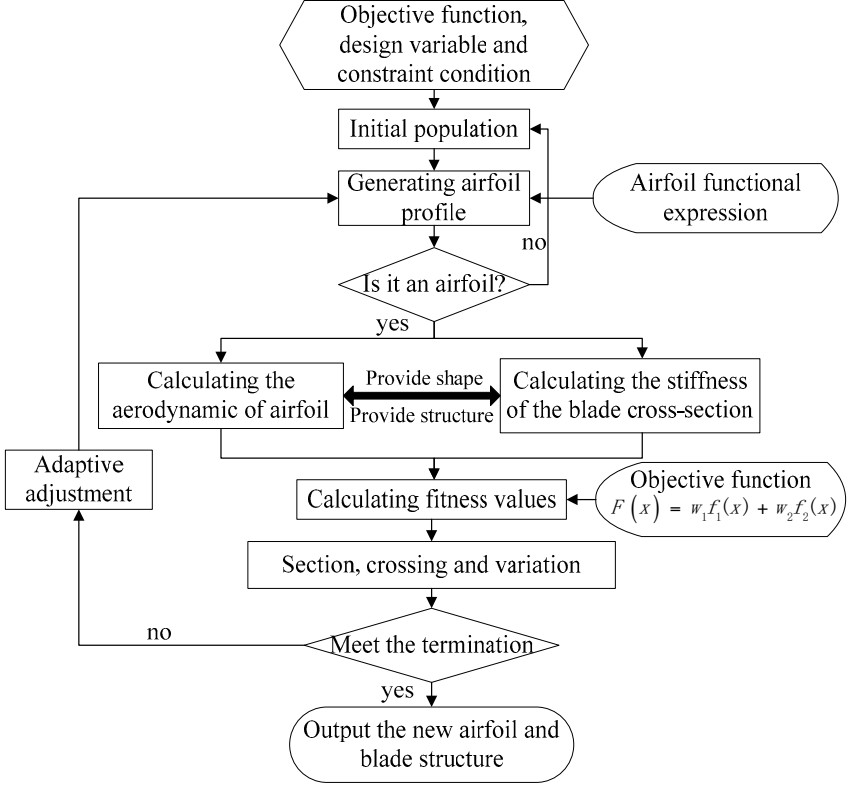

**Figure 5.** Flow chart of the optimal design method for the airfoil and its structure.

## 6. Optimal Results and Discussion

Two new airfoils with the maximum relative thickness of about 30% are optimized by the above two schemes. The airfoil designed by scheme 1 is named WQ-A300, and the airfoil designed by scheme 2 is named WQ-B300. The profiles of the new airfoils and the classic DU97-W-300 airfoil which was designed by Delft University are plotted in Figure 6.

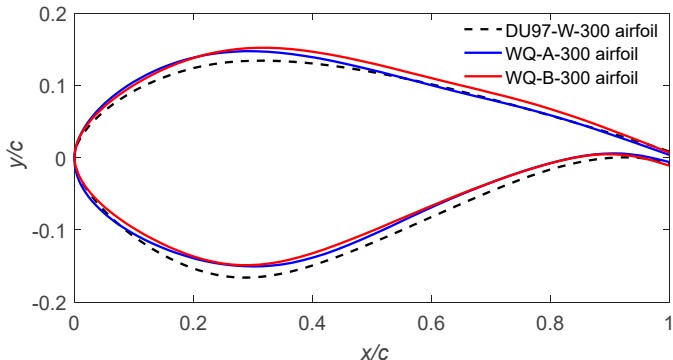

**Figure 6.** Profiles of the new airfoils and the DU97-W-300 airfoil.

To verify the superiority of the aerodynamic characteristics of the optimized airfoils, the aerodynamic performance of the new airfoils and the DU97-W-300 airfoil were compared. Figures 7 and 8 show a comparison of the aerodynamic performance of the new airfoils with that of the DU97-W-300 airfoil (Reynolds number Re = $3.0 \times 10^6$, Mach number Ma = 0.15). The key aerodynamic parameters of the WQ-A300, WQ-B300, and DU97-W-300 airfoils are listed in Table 4. The aerodynamic characteristics of wind turbine airfoils are calculated by the RFOIL software. The accuracy and reliability of the theoretical values calculated by the software have been verified in some relevant studies [3,4]. According to the figures and table, compared to the DU97-W-300 airfoil, the WQ-A300 and WQ-B300 airfoils exhibit a better lift coefficient and lift-drag ratio under both smooth and rough conditions. Under the smooth condition, the maximum lift coefficient and lift-drag ratio of the WQ-A300 airfoil are 1.889 and 141.756 respectively, an increase by 15.38% and 15.6%, and the maximum lift coefficient and lift-drag ratio of the WQ-B300 airfoil are 1.823 and 139.647 respectively, an increase of 11.8%, and 13.9%. Under the rough condition, the maximum lift coefficient and lift-drag ratio of the WQ-A300 airfoil are 1.326 and 66.391 respectively, an improvement by 4.57% and 7.27%, and the maximum lift coefficient and lift-drag ratio of the WQ-B300 airfoil are 1.295 and 64.798 respectively, an improvement of 2.1%, and 4.7%.

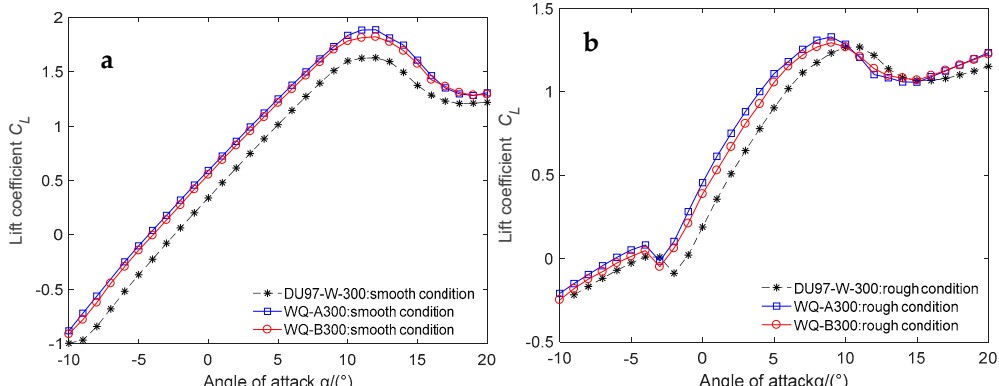

**Figure 7.** (**a**) Lift coefficients of the new airfoils and the DU97-W-300 airfoil (smooth condition); (**b**) Lift coefficients of the new airfoils and the DU97-W-300 airfoil (rough condition).

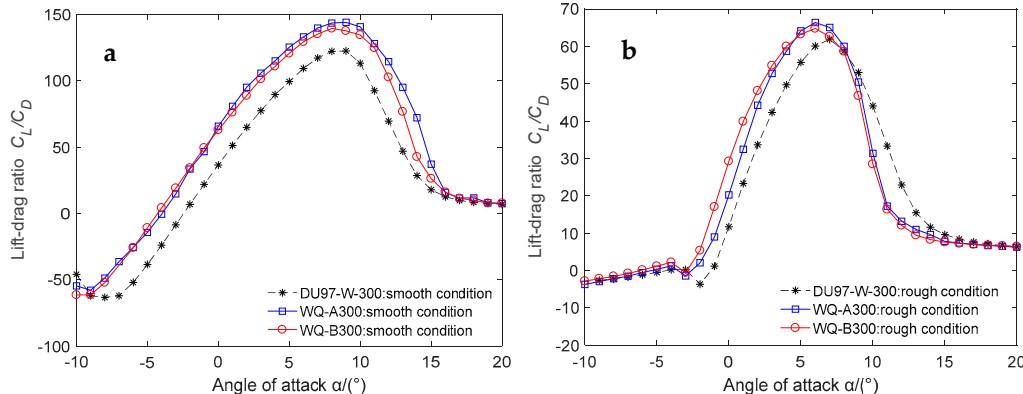

**Figure 8.** (**a**) Lift-drag ratio of the new airfoils and the DU97-W-300 airfoil (smooth condition); (**b**) Lift-drag ratio of the new airfoils and the DU97-W-300 airfoil (rough condition).

**Table 4.** The comparisons of the key aerodynamic performance.

| Airfoil Name | Smooth Condition | | Rough Condition | |
|---|---|---|---|---|
| | $C_{L,max}$ | $L/D_{,max}$ | $C_{L,max}$ | $L/D_{,max}$ |
| DU97-W-300 | 1.631 (12°) | 122.638 (9°) | 1.268 (11°) | 61.889 (7°) |
| Reference [32] airfoil | 1.801 (12°) | 128.449 (8°) | 1.292 (9°) | 63.176 (6°) |
| WQ-A300 | 1.889 (14°) | 141.756 (9°) | 1.326 (10°) | 66.391 (6°) |
| WQ-B300 | 1.823 (12°) | 139.647 (8°) | 1.295 (9°) | 64.798 (6°) |

Note: The location of angle of attack is in brackets; $C_{L,max}$ is the maximum lift coefficient; and $L/D_{,max}$ is maximum lift-drag ratio.

The profiles of the airfoil with a 30% thickness in literature [32] and the WQ-B300 airfoil are plotted in Figure 9. The aerodynamic characteristic of the WQ-B300 airfoil is compared with that of the airfoil in literature [32] (as shown in Figures 10 and 11). Table 4 lists the key aerodynamic parameters. Compared with that of the airfoil in literature [32], the maximum lift-drag ratio $C_L/C_D$ of the WQ-B300 airfoil is 139.647, increased by 8.718% under the smooth condition. The other aerodynamic parameters are more or less the same with that of the airfoil in literature [32].

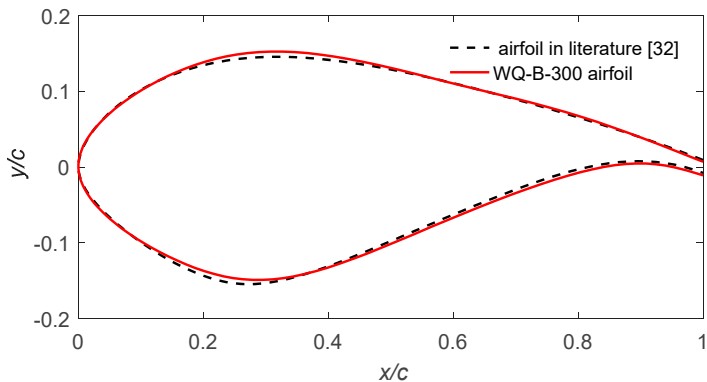

**Figure 9.** Profiles of the airfoil in literature [32] and the new airfoil.

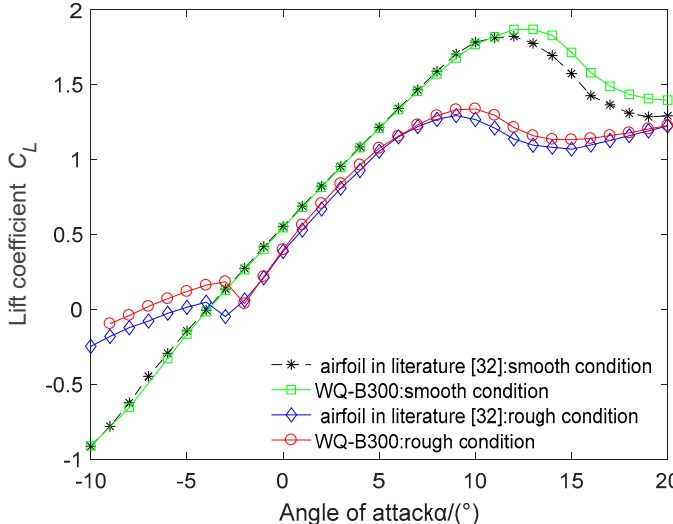

**Figure 10.** Lift coefficients of the airfoil in literature [32] and the WQ-B300 airfoil.

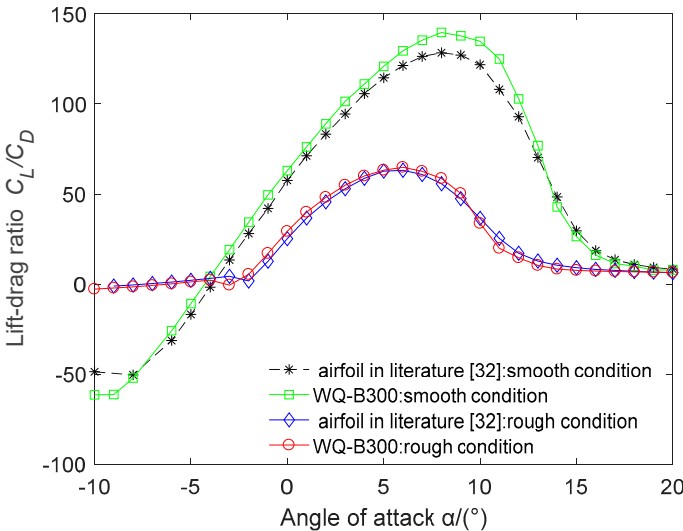

**Figure 11.** Lift-drag ratio of the airfoil in literature [32] and the WQ-B300 airfoil.

The comparison of the cross-sectional plying parameters of the WQ-A300 airfoil and the WQ-B300 airfoil is shown in Figure 12. The variety of the main beam location of the WQ-B300 cross-section is shown in Figure 13. It should be explained that the original structural parameters of the new blade cross-section are from the data of the 2.1 MW wind turbine blade (see Tables 1 and 2 for details). The cross-sectional plying parameters of the WQ-A300 airfoil is the same as that of the DU97-W-300 airfoil. Table 5 gives the comparisons of the cross-sectional layer parameters of the WQ-A300 airfoil and the WQ-B300 airfoil. From the above figures and table, we can see that the thickness of the uniaxial lamination of the WQ-B300 cross-section is decreased and the thickness of the biaxial and triaxial lamination is increased by optimizing the layer parameters of the blade cross-section and the position of the main beam. In addition, the number of uniaxial layers at the leading and trailing edges of the WQ-B300 is decreased, while the number of other layers is added correspondingly. At the same time, the position of the main beam of the WQ-B300 airfoil cross-section is moved forward accordingly, as shown in Figure 13. As a result, the optimization of the layer parameters makes the WQ-B300 blade cross-sectional stiffness increased correspondingly, as shown in Table 6. Compared with the stiffness of the blade cross-section with the DU97-W-300 airfoil, the flapwise and edgewise stiffness of the blade cross-section of the WQ-B300 airfoil are $1.19 \times 10^9$ and $3.87 \times 10^8$, increased by 6.2% and

8.4%, respectively. However, the flapwise stiffness of the blade cross-section with the WQ-A300 airfoil increased slightly, and the edgewise stiffness decreased by 2.5%. The main reason is that the WQ-B300 airfoil and its blade cross-section are designed by changing both the airfoil geometry and blade laminated parameters, such that the flapwise and edgewise stiffness were improved. Nevertheless, in the optimal process of the WQ-A300 airfoil and its blade cross-section, the blade stiffness is only affected by changing the airfoil geometry. Although the aerodynamic performance of the WQ-A300 airfoil increased obviously, the blade cross-section flapwise and edgewise stiffness increased slightly, or even decreased partly.

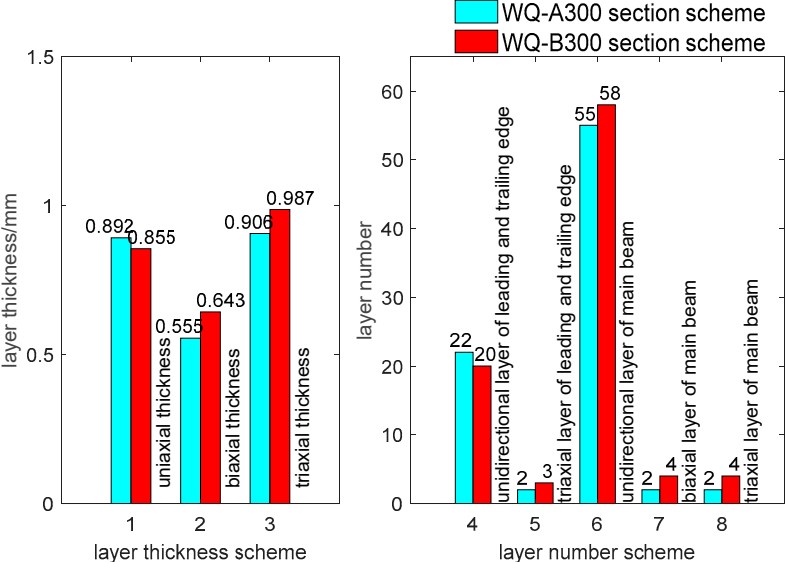

**Figure 12.** Comparisons of layer parameters for the WQ-A300 and WQ-B300 blade sections.

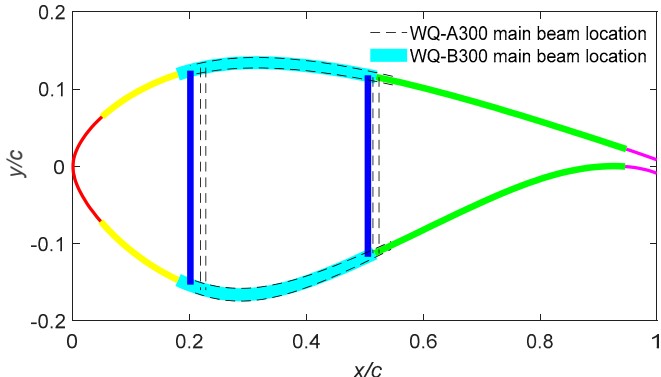

**Figure 13.** Variation of the main beam position for the WQ-B300 blade cross-section.

**Table 5.** Comparisons of the layer parameters of the DU97-W-300 and WQ-B300 blades.

| Layer Parameters | DU97-W-300 Section | WQ-B300 Section |
|---|---|---|
| Uniaxial thickness (mm) | 0.892 | 0.855 |
| Biaxial thickness (mm) | 0.555 | 0.643 |
| Triaxial thickness (mm) | 0.906 | 0.987 |
| Uniaxial layer of leading and trailing edge | 22 | 20 |
| Triaxial layer of leading and trailing edge | 2 | 3 |
| Uniaxial layer of main beam | 55 | 58 |
| biaxial layer of main beam | 2 | 4 |
| triaxial layer of main beam | 2 | 4 |
| Main beam position | 0.20 C<br>0.55 C | 0.185 C<br>0.522 C |

**Table 6.** Stiffness comparisons of the new section and the DU97-W-300 blade section.

| Stiffness | DU97-W-300 Section | WQ-A300 Section | WQ-B300 Section |
|---|---|---|---|
| Flapwise $EI_x$ (N·m$^2$) | $1.12 \times 10^9$ | $1.13 \times 10^9$ | $1.19 \times 10^9$ |
| Edgewise $EI_y$ (N·m$^2$) | $3.57 \times 10^8$ | $3.48 \times 10^8$ | $3.87 \times 10^8$ |

## 7. Conclusions

For the parametric design of the medium thickness airfoils, an integrated design method of wind turbine airfoil based on aerodynamic performance and blade structural stiffness is proposed in this work. A multi-objective optimal mathematic model with the objective functions of aerodynamic performance and blade cross-section stiffness is established. A new WQ-B300 airfoil with high aerodynamic and cross-section stiffness performance was optimized by using a genetic algorithm coupled with the functional integrated expression of airfoil and the calculation program of blade cross-sectional stiffness matrix.

Then, the aerodynamic performance and structural stiffness characteristics of the WQ-B300 airfoil were compared to those of the DU97-W-300 and WQ-A300 airfoils. Compared with that of the DU97-W-300 airfoil, the maximum lift coefficient of the WQ-B300 airfoil is 1.823 and 1.295, increased by 11.8% and 2.1%, respectively, under the smooth condition and rough conditions. The maximum lift-drag ratio of the WQ-B300 airfoil is 139.647 and 64.798, increased by 13.9% and 4.7%, respectively, under the smooth condition and rough conditions. In addition, compared with the airfoil with 30% thickness in literature [32], the aerodynamic performance of the WQ-B300 airfoil exhibits a higher maximum lift-drag ratio in the smooth condition, which is increased by 8.7%. Compared with that of the DU97-W-300 airfoil, the blade cross-sectional flapwise and edgewise stiffness is increased by 6.2% and 8.4%, respectively. The main reason is that the aerodynamic and stiffness characteristics of the WQ-B300 airfoil are taken into account in the process of optimal design. However, the WQ-A300 airfoil is designed mainly considering the aerodynamic performance improvement.

In summary, the traditional design of airfoils with the medium thickness (the maximum relative thickness is 25%~35%) often neglects the structural characteristics of the blades and focuses on the improvement of aerodynamic performance. However, the influence of the blade cross-sectional stiffness is ignored. For the design of the medium thickness airfoils, not only the aerodynamic performance of the blade should be considered to improve the utilization of wind energy, but also the stiffness performance of the blade should be taken into account to improve the strength and stability of the blade.

**Author Contributions:** Conceptualization, Q.W. and J.W.; methodology, Q.W.; software, P.H.; validation, Q.W., P.H. and D.G.; formal analysis, Q.W.; investigation, P.H.; resources, Q.W.; data curation, D.G.; writing—original draft preparation, Q.W.; writing—review and editing, J.W.; visualization, P.H.; supervision, D.G.; project administration, Q.W.; funding acquisition, Q.W.

**Funding:** This research was funded by National Natural Science Foundation of China, grant number 51975190 and 51405140.

**Conflicts of Interest:** The authors declare no conflict of interest.

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
