# Peer review of "Integrated Design of Aerodynamic Performance and Structural Characteristics for Medium Thickness Wind Turbine Airfoil"

_applsci, doi:10.3390/app9235243_

Round 1
Reviewer 1 Report
This manuscript has been significantly improved, and I recommend acceptance under minor revisions.
During the last round of revisions, the two reviewers warned the authors concerning some pieces of the manuscript with different font sizes. The authors did not fix this mistake, and honestly this is a primary mistake. The article MUST not be published with different font sizes throughout the text. This will make the article look like a disaster. I am still missing a discussion in more depth comparing your results with results from the literature. This is what I recommended in the last round of revisions, and I did not see any improvements in this regard.
Author Response
Dear Editors and Reviewers,
Thank you for your very useful comments and suggestions on the contents of our manuscript. We have modified the manuscript accordingly, and detailed corrections are listed below point by point:
For Reviewer #1:
(1) During the last round of revisions, the two reviewers warned the authors concerning some pieces of the manuscript with different font sizes. The authors did not fix this mistake, and honestly this is a primary mistake. The article MUST not be published with different font sizes throughout the text. This will make the article look like a disaster. I am still missing a discussion in more depth comparing your results with results from the literature. This is what I recommended in the last round of revisions, and I did not see any improvements in this regard.
Answer: we have changed the words with the same font sizes (the blue words in our paper). In fact, we have compared our results of the new airfoils with results from DU97-W-300 airfoil and from the airfoil in literature [31]. As for the comparison of the structural characteristic, we have explained our reason in the first revision.
Thank you again for your constructive comments and suggestions. Communicating with the reviewers and editors, we have learned a lot about this study. The manuscript has been resubmitted to your journal. We look forward to your positive response.
Sincerely,
Quan Wang
School of Mechanical Engineering
Hubei University of Technology
WuHan
China
Email: wangquan2013@hbut.edu.cn; quan_wang2003@163.com

Reviewer 2 Report
1. line 209: refence quotation of "PreComp software" is missing
2. quote "RFOIL" in line 103 or in line 226. not omly in line 283
Author Response
Dear Editors and Reviewers,
Thank you for your very useful comments and suggestions on the contents of our manuscript. We have modified the manuscript accordingly, and detailed corrections are listed below point by point:
For Reviewer #2:
(1) Line 209: reference quotation of "PreComp software" is missing.
Answer: we have checked it again and added the reference quotation of “PreComp” in line 223 (the green words).
(2) Quote "RFOIL" in line 103 or in line 226. not only in line 283
Answer: we have checked it again and added the reference quotation of “RFOIL” in line 111, in line 223 and 241 (the green words).
Thank you again for your constructive comments and suggestions. Communicating with the reviewers and editors, we have learned a lot about this study. The manuscript has been resubmitted to your journal. We look forward to your positive response.
Sincerely,
Quan Wang
School of Mechanical Engineering
Hubei University of Technology
WuHan
China
Email: wangquan2013@hbut.edu.cn; quan_wang2003@163.com

Reviewer 3 Report
This is an interesting paper dealing with a topical subject. Generally the structure of the paper is adequate to the function with a number of well-presented figures and tables. It would be useful if the references to authors' names employed a consistent approach eg line 33 lower case, line 43 upper case. Line 110 integrated express? Line 161 Poisson's ratio. Line 246 Scheme 1 Line 249 Scheme 2 Line 290 Why was the population chosen as 30 together with other optimisation parameters? Line 310 What are the roughness values and are they consistent between your cases and the Delft profile? Line 331 In order to validate the new airfoils exhibit higher dynamic performance? What is the meaning of this sentence? Line 411 Some of the names are written in full capitals (424 etc) whilst most are not.
Author Response
Dear Editors and Reviewers,
Thank you for your very useful comments and suggestions on the contents of our manuscript. We have modified the manuscript accordingly, and detailed corrections are listed below point by point:
For Reviewer #3:
(1) This is an interesting paper dealing with a topical subject. Generally the structure of the paper is adequate to the function with a number of well-presented figures and tables. It would be useful if the references to authors' names employed a consistent approach eg line 33 lower case, line 43 upper case. Line 110 integrated express? Line 161 Poisson's ratio. Line 246 Scheme 1 Line 249 Scheme 2.
Answer: we have checked them again and changed the problems of the formats and some spelling mistakes (the red words).
(2) Line 290 Why was the population chosen as 30 together with other optimization parameters?
Answer: through adjusting the GA parameters, we find that these optimized parameters are appropriate during the optimization process for wind turbine airfoils.
(3) Line 310 What are the roughness values and are they consistent between your cases and the Delft profile?
Answer: The free transition model is used to simulate the smooth surface flow, while the fixed transition model is used to simulate the roughness surface flow (the transition points for the suction and pressure sides are fixed at the chord of 1% and 10%, respectively). They are consistent between our cases and the Delft profile.
(4) Line 331 In order to validate the new airfoils exhibit higher dynamic performance? What is the meaning of this sentence?
Answer: we have deleted this sentence.
(5) Line 411 Some of the names are written in full capitals (424 etc) whilst most are not.
Answer: we have checked them again and changed them (the red words).
Thank you again for your constructive comments and suggestions. Communicating with the reviewers and editors, we have learned a lot about this study. The manuscript has been resubmitted to your journal. We look forward to your positive response.
Sincerely,
Quan Wang
School of Mechanical Engineering
Hubei University of Technology
WuHan
China
Email: wangquan2013@hbut.edu.cn; quan_wang2003@163.com

This manuscript is a resubmission of an earlier submission. The following is a list of the peer review reports and author responses from that submission.
Round 1
Reviewer 1 Report
This manuscript presents a method to improve performance of an existing medium-thickness airfoil, aiming to optimize the lift coefficient and the lift/drag ratio while considering structural stiffness characteristics. The authors developed their own workflow to approach the problem, and at the end they were able to reach an interesting level of improvement (around 15%) in terms of aerodynamic performance. How is this level of improvement compared with other similar works in literature? The authors never discussed it in the results section. Therefore, it is hard to tell how successful they were at their attempt, since there is no discussion of any sort in this manuscript. The authors did not compare their results against similar findings in literature.
The authors need to improve the introduction. There are many previous works in literature in the same research topic of this manuscript, but the authors did a poor job on describing existing literature and pointing out existing gaps/possibilities for improvements. They described only a few previous works, and in my view, they could have done a much better job.
In regard to the methods section, I personally do not have problems and I think they did a good job. They were careful to include equations, assumptions, etc.
The results section is also good, but as I said they did not write a discussion against other similar findings in literature. The results were only presented, but not discussed in depth.
The conclusions are consequently poor. For instance, they say at the conclusions that a significant level of improvement has been achieved. How do you prove it is significant if you haven’t even discussed your findings against other similar findings?
But what I am mostly concerned with is the writing style/grammar. I found so many mistakes regarding grammar/writing style that I think the manuscript must go under extensive reviews. Honestly, I am inclined to reject the manuscript in the next round of revisions if the authors do not extensively review grammar mistakes/writing style. I did my best to point out some of the mistakes (please find the pdf file that I attached), but I am sure that there can be others.
From the point of view of the scientific soundness of the manuscript, it is definitely acceptable. From the point of view of the way the manuscript is currently presented, it is unacceptable.

Reviewer 2 Report
The English must be deeply improved as well as the format of the manuscript. At its current status, it is hardly readable and understandable. Besides, the format does not meet journal standards. In addition to it, the work is not well explained and then even though the format, style, and English were appropriate, it will not be easy to understand.
Example of style improvements: 1) the authors are using two or three different fonts and/or sizes for the text. It looks like it is a copy-paste of the writing from another document. 2) The sections & subsections are not clearly separated. Their name does not start with capital letters...
Reference section: inconsistencies in the style between different references.
Content: Subsection 1.Airfoils with medium thickness functional integrated express -> This part is standard, it does not represent a new contribution of the authors.
In summary, the work might be of interest to the industry and/or academic community, but in its form it is impossible to be understood and therefore it will not be valuable.